# Sleep disrupts complex spiking dynamics in the neocortex and hippocampus

**Joaquín González**[1,2], **Matias Cavelli**[3], **Adriano B. L. Tort**[2], **Pablo Torterolo**[1], **Nicolás Rubido**[4,5]*

**1** Departamento de Fisiología de Facultad de Medicina, Universidad de la República, Montevideo, Uruguay, **2** Brain Institute, Federal University of Rio Grande do Norte, Natal, Brazil, **3** Department of Psychiatry, University of Wisconsin, Madison, Wisconsin, United States of America, **4** University of Aberdeen, King's College, Institute for Complex Systems and Mathematical Biology, Aberdeen, United Kingdom, **5** Instituto de Física, Facultad de Ciencias, Universidad de la República, Montevideo, Uruguay

* nicolas.rubidoobrer@abdn.ac.uk, nrubido@fisica.edu.uy

**Data Availability Statement:** All relevant data for this study are publicly available from the CRCNS databases (https://crcns.org/data-sets/fcx/fcx-1 and https://crcns.org/data-sets/hc/hc-11).

## Abstract

Neuronal interactions give rise to complex dynamics in cortical networks, often described in terms of the diversity of activity patterns observed in a neural signal. Interestingly, the complexity of spontaneous electroencephalographic signals decreases during slow-wave sleep (SWS); however, the underlying neural mechanisms remain elusive. Here, we analyse *in-vivo* recordings from neocortical and hippocampal neuronal populations in rats and show that the complexity decrease is due to the emergence of synchronous neuronal DOWN states. Namely, we find that DOWN states during SWS force the population activity to be more recurrent, deterministic, and less random than during REM sleep or wakefulness, which, in turn, leads to less complex field recordings. Importantly, when we exclude DOWN states from the analysis, the recordings during wakefulness and sleep become indistinguishable: the spiking activity in all the states collapses to a common scaling. We complement these results by implementing a critical branching model of the cortex, which shows that inducing DOWN states to only a percentage of neurons is enough to generate a decrease in complexity that replicates SWS.

## Introduction

Cognition and behaviour drastically change across the sleep-wake cycle [1]. During wakefulness, animals are able to interact with their environment, but lose this ability as they fall asleep. During sleep, there is an alternation between slow-wave sleep (SWS), associated with diminished cognitive capacities, and rapid eye movement (REM) sleep, an active state where most dreams occur [2, 3]. The electroencephalogram (EEG) concomitantly changes along with behavior: fast and desynchronised activity appears during wakefulness and REM sleep, while slow quasi-synchronous patterns characterize SWS. Nevertheless, in spite of having well-documented, state-dependent EEG signatures, their underlying mechanisms remain to be fully understood.

In the last decade, there has been a significant rise in the use of complexity metrics (which often measure the diversity of patterns in a signal) capable of revealing hidden non-linear

**Funding:** J.G acknowledges the support of Comisión Académica de Posgrado (CAP), CSIC Iniciación and PEDECIBA. P.T also acknowledges the support of PEDECIBA. A.B.L.T acknowledges the support of CAPES and CNPq. N.R. acknowledges the CSIC group grant "CSIC2018 - FID 13 - Grupo ID 722. The funders had no role in study design, data collection and analysis, decision to publish, or preparation of the manuscript.

**Competing interests:** The authors have declared that no competing interests exist.

effects in electrophysiological recordings. These tools have repeatedly shown that the complexity of EEG signals decreases during unconscious states, such as during sleep [4–15] or anaesthesia [14, 16–23]. However, these macroscopic signals have major limitations: they tend to be contaminated by confounding variables (e.g., muscular activity or eye movements) and recovering their exact neural source is often impossible. Thus, the neural patterns driving complexity changes across the sleep-wake states have not been elucidated.

A possible mechanism causing the complexity reduction during sleep is the emergence of DOWN states, defined as synchronous periods of spiking silence [24–29] which generate the extracellular slow waves characteristic of SWS [24–31]. These states are hypothesised to disrupt neural interactions [32], and have been shown to directly alter the complexity of evoked cortical responses [33, 34]. However, no direct analysis of *in-vivo* neuronal populations has shown that DOWN states reduce the complexity of spontaneous cortical activity during sleep.

Here, we analyse *in-vivo* recordings of neuronal populations in the neocortex and hippocampus, quantifying their spontaneous ensemble dynamics in terms of their phase-space recurrences. Our analyses, along with neuronal modelling, show that DOWN states fully account for the complexity decrease during SWS, while a common spiking regime characterises all sleep-wake states in the neocortex and hippocampus.

## Results

We study *in-vivo* neuronal recordings from the neocortex and hippocampus of 15 rats cycling through the states of wakefulness (Wake), slow-wave sleep (SWS), and rapid-eye movement (REM) sleep (Fig 2**A**). We analyse ≃1600 neurons, corresponding to 31 independent sessions with 51±5 neurons simultaneously recorded (details in Methods: Datasets). We use recurrence quantification analysis (RQA) to characterise the evolution of the whole population firing counts in each session during each sleep-wake state, extending the characterisation of a population activity beyond a single measure (such as Hurst exponent, entropy, or fractal dimension) or an aggregate of individual neurons. We complement the RQA with coherence and entropy analyses of local field potentials (LFP), spike avalanches, and a critical branching model.

### Recurrence analysis reduces high-dimensional dynamics to a 2D representation

The population activity from a cortical location at any given time is a high-dimensional variable detailing the system instantaneous state, i.e., the spiking activity of all neurons (Fig 1**A**, **left**). Its evolution gives a trajectory in the $N$-dimensional phase-space, which has the firing counts of each neuron as its components (Fig 1**A**, **right**). An attractor is evidenced as a manifold that attracts different trajectories of the system to the same region of the phase-space; the more convoluted (fractal) the attractor is, the higher the temporal complexity of its trajectories. The trajectory of a cortical area is typically high-dimensional, since 50 neurons from any given experimental session results in $N = 50$-dimensional phase-space. By applying Recurrence Quantification Analysis (RQA), we reduce these dimensions to the analysis of 2-dimensional recurrence plots (RP) (Fig 1**C**).

We construct a recurrence plot as follows. Let $\{\vec{x}(t_1), \vec{x}(t_2), \ldots, \vec{x}(t_n)\}$ be a trajectory, where $\vec{x}(t_i)$ is the state-vector whose components are firing counts, $x_k(t_i)$, for each neuron $k$ in the population ($k = 1, \ldots, N$) at time $t_i$ with $i = 1, \ldots, T$, $T$ being the number of 50 *ms* time bins. We choose this time-bin width to match the definition of a neocortical OFF-period, i.e., a period $\geq 50$ *ms* without spikes. Hence, our firing counts are integer variables that can range from 0 up to 50 (assuming a maximum of 1 spike per *ms*). A recurrence plot is then defined by a symmetric matrix whose entries are $R(i, j) = 1$ if $\|\vec{x}(t_i) - \vec{x}(t_j)\| < \epsilon$, or $R(i, j) = 0$ otherwise,

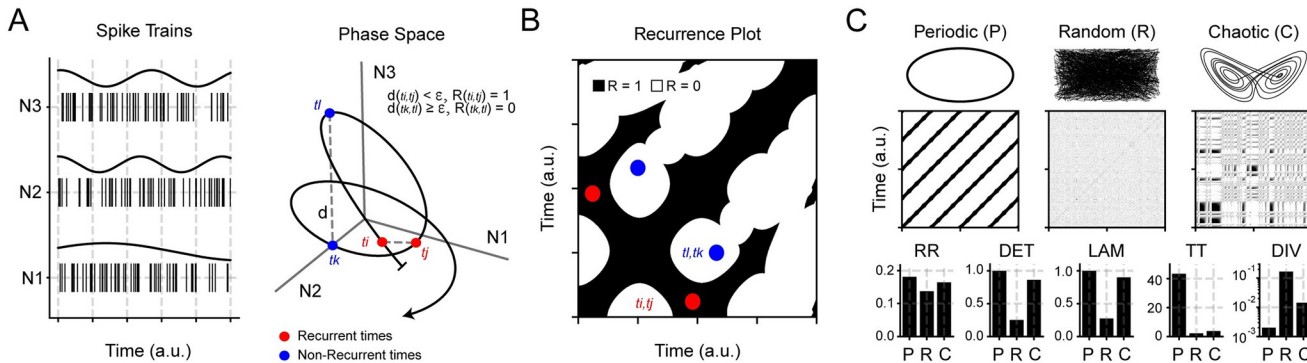

**Fig 1. Recurrence example of population activity. A Left** Example of spike trains for 3 neurons (N1-N3). The continuous line on top shows the firing counts of each spike train. **Right** Resultant phase-space trajectory (evolution), where the axes represent the firing counts of each neuron. For every pair of points in the trajectory, their distance (d) is computed (the dashed lines illustrate two such distances); If the distance is less than a predefined $\epsilon$ value, a recurrence between the time points is defined to occur. Two recurrent times are shown in red (ti,tj), while two non-recurrent times are shown in blue (tk, tl). **B** Recurrence plot for the trajectory shown in panel **A**. Red and blue time pairs are now depicted as coordinates in the resulting map. **C** Example recurrence plots from periodic, random, and chaotic trajectories. On top of the recurrence plot, we show the phase-space of the example; below we illustrate how the RQA metrics behave for each trajectory type. RR: Recurrence Rate, DET: Determinism, LAM: Laminarity, TT: Trapping Time, DIV: Divergence.

with $i, j = 1, \ldots, T$ and $\epsilon > 0$ defining closeness. A recurrence happens whenever the system trajectory returns to the same region of phase-space up to $\epsilon$. We set $\epsilon = \sigma_p$ ($\sigma_p$ being the standard deviation of the population activity during wakefulness) to guarantee a sufficiently sparse plot but with sufficient points to carry statistical analyses. Nevertheless, our results are robust to changes in $\epsilon$ or time-bin width (S1 Fig).

Two generic structures appear in a recurrence plot: diagonal lines, originating from periodic trajectories, and vertical lines, originating from trapped (frozen) trajectories. These structures help to differentiate between periodic, random, or chaotic trajectories (corresponding panels in Fig 1**C**), which can be quantified by different metrics (see RQA in Methods). We use RQA to measure (i) Recurrence Rate (density of points), RR, (ii) Determinism (proportion of points forming diagonal lines), DET, (iii) Laminarity (proportion of points forming vertical lines), LAM, (iv) Trapping Time (average length of vertical lines), TT, and (v) Divergence (inverse of the longest diagonal line, excluding the identity line), DIV.

To illustrate how the RQA metrics behave, we compute them for the examples of Fig 1**C**. RR is slightly larger for the periodic system since it recurs more often into similar states than the other examples, while the chaotic trajectory recurs more than the random example. DET and LAM, on the other hand, are maximal for periodic and chaotic systems because all points form vertical and diagonal structures, while these drop near zero for the random system since recurrent times are rarely connected. Moreover, TT is larger for the periodic system since there are no isolated recurrent times (all points form small vertical structures). TT decreases in the chaotic system due to isolated recurrent times and lowers even further for the random system because recurrences occur by chance and rarely form any vertical structure. Finally, DIV is the largest for the random system since no diagonal structures are formed, while DIV plummets to near zero for the periodic system since all points form long diagonal lines. DIV lies in-between for the chaotic system since it forms short diagonal lines. Thus, predictability in the system trajectory is quantified by RR, DET, LAM, and TT, where the larger [smaller] their values, the more [less] predictable. On the other hand, randomness is quantified by DIV, where the larger [smaller] its value, the more divergent [convergent] the trajectory.

## The complexity of neuronal dynamics is reduced during slow-wave sleep

Fig 2**A** shows the LFP and spike trains of frontal cortex neurons in a session for a representative animal under each sleep-wake state. The corresponding recurrence plots for 10-second trajectories are shown in Fig 2**B**. Note that SWS exhibits a denser plot than Wake or REM sleep, implying that SWS has firing patterns recurring more often than Wake or REM sleep. Also, SWS shows a distinctive square-shaped recurrence pattern, which points to the existence of time windows when the trajectory of the population activity is frozen (or practically unchanged). The RQA metrics applied to all available 10-second trajectories for all recorded sessions confirm that frontal cortex activity (~900 neurons in total) is significantly more predictable and less random during SWS than Wake or REM sleep (Fig 2**C**; see statistics in S1 Table in S1 File).

Specifically, SWS has the largest RR, DET, LAM, and TT, indicating high predictability of the neuronal activity, whereas it has the smallest DIV, suggesting that SWS is less random than Wake or REM sleep. Noteworthy, these RQA changes during SWS correlate with the number of recorded neurons (S2 Fig), suggesting the complexity reduction is a population-level phenomena. Also note that the RQA differences across states are not due to a change of the attractor's topology (S3 Fig). Moreover, the RQA results hold true when dividing the frontal cortex into specific areas (Fig 2**D** and 2**E**) or when analysing the population activity from the hippocampus (S2 Table in S1 File). In fact, when comparing the RQA metrics among the secondary

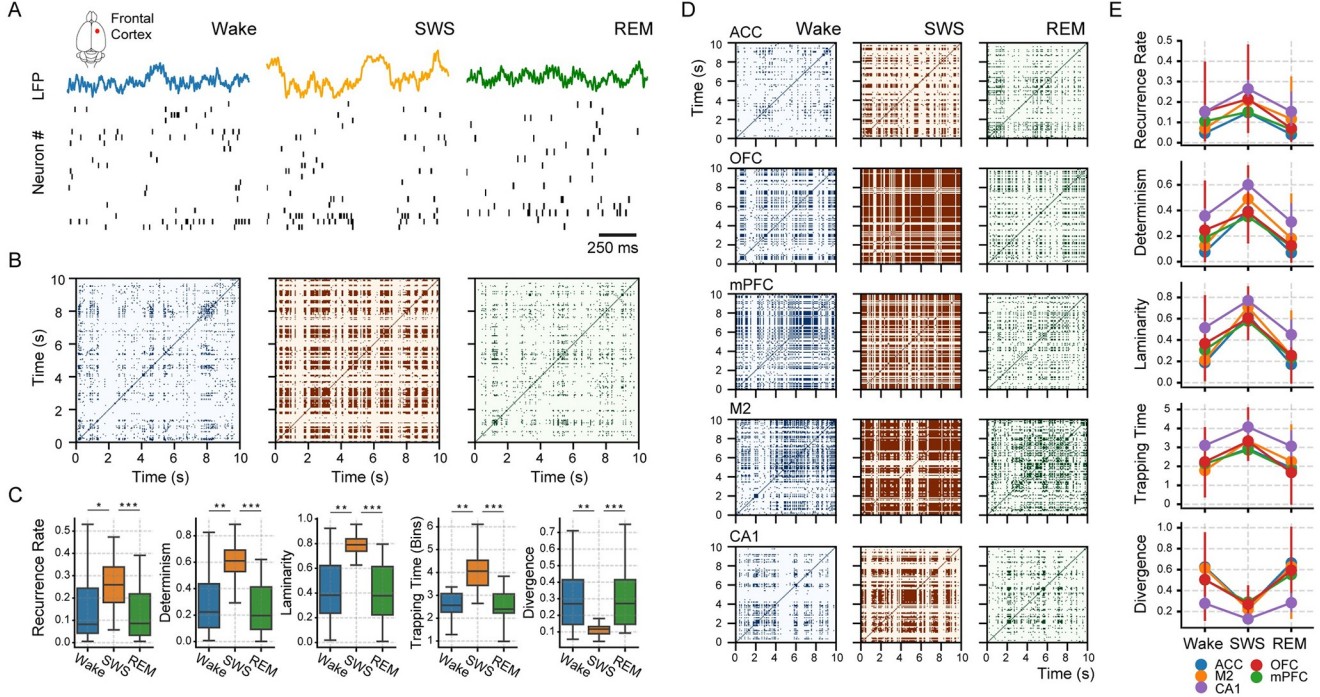

**Fig 2. Recurrence quantification analysis (RQA) of *in- vivo* population activity from the frontal cortex and hippocampus. A** Local field potentials (LFP) and spike-train raster plots (1*s* interval) for a representative rat during Wake (left), SWS (middle), and REM sleep (right). **B** Recurrence plots constructed from a 10*s* interval of the population activity (see Methods for details). **C** 5 RQA metrics for the sleep-wake states; boxplots show results from the pool of 24 sessions across 12 animals (outliers are not shown). *$p < 0.001$, **$p < 0.0001$, ***$p < 0.00001$ (corrected for multiple comparisons). **D** Example recurrence plots for different cortical locations sleep-wake states. ACC: anterior cingulate cortex; OFC: orbito-frontal cortex; mPFC: medial prefrontal cortex; M2: secondary motor cortex; CA1: hippocampus. **E** RQA metrics for the sleep-wake states in each cortical area shown in the previous panel.

motor cortex (M2), medial prefrontal cortex (mPFC), orbitofrontal cortex (OFC), anterior cingulate cortex (ACC) and the CA1 hippocampus region, we find no statistical differences (Kruskal-Wallis test across cortex: RR [$p$ = 0.68], DET [$p$ = 0.39], LAM [$p$ = 0.69], TT [$p$ = 0.21] and DIV [$p$ = 0.46]), suggesting that the complexity reduction in the spiking activity is conserved across brain regions. Thus, these results demonstrate that SWS is the least complex spiking state, consistent with previous reports of decreased EEG complexity during sleep [4–15] or anaesthesia [14, 16–22].

## Neuronal recurrences during SWS are mainly driven by DOWN states

The square-shaped recurrences appearing during SWS in Fig 2 can be generated by two possible mechanisms. Either a subset of the neurons (or even all) remains constantly active for a period of time, or the neurons remain silent (null firing counts) corresponding to a trajectory in the origin of the phase space. Next, we show that the latter is true and is mainly due to DOWN states.

DOWN states are the neural substrate underlying slow-wave activity (0–4$Hz$) [24–26, 28, 30]. Therefore, a correlation between recurrent trajectories and DOWN states provides a physiological mechanism for the loss of complexity during sleep. For the neocortex, DOWN states can be obtained by finding OFF- periods [26, 28, 29], i.e., periods $\geq 50ms$ when almost all neurons remain silent [26] (Fig 3A left). For the hippocampus, we obtain DOWN states by selecting the times when less than 10% of the recorded neurons fired since hippocampal neurons maintain a minimal firing activity during DOWN states [35] (Fig 3A right).

The time-series of DOWN states in the neocortex match the times when the trajectory has a recurrence (see the pink and black curves in Fig 3B bottom panel). We find a significant correlation between these time-series ($R = 0.77 \pm 0.02$, $p < 10^{-64}$ for all sessions). This means that the majority of the SWS recurrences is due to DOWN states. We observe a similar scenario in the hippocampus (right panels in Fig 3A and 3B), where we find an even higher correlation between the time-series of DOWN states and that of the recurrences ($R = 0.84 \pm 0.01$, $p < 10^{-64}$ for all sessions).

Notably, SWS becomes more complex if we exclude DOWN states; namely, if we employ a trajectory containing only the population UP states. The corresponding recurrences and metrics are shown in Fig 3D and 3E; note that RR, DET, LAM, TT and DIV significantly change and become comparable to those of Wake and REM sleep (S4 Fig and Fig 2C). Overall, these results support the hypothesis that neuronal trajectories are similar in SWS UP states to Wake and REM sleep [36, 37].

## Neocortical DOWN states explain the EEG complexity reduction during sleep

We next investigate how the population activity results translate to the EEG and LFP signals simultaneously recorded from the freely moving animals. To that end, we create synthetic local field potentials (sLFP) (Fig 4) from the actual excitatory spiking activity, in which we assume that each spike generates an exponentially decaying PSP. The motivation behind this method is that it allows to precisely control the sources which dictate the field potential and avoid the influence of any external variable not directly related to spiking activity (such as EMG contamination and volume-conducted signals [38]). Since LFPs primarily reflect postsynaptic potentials (PSPs) [39], we average the modelled PSPs over the population of neurons at each time in order to obtain the instantaneous sLFP (Fig 4A).

We find that sLFPs have asynchronous low-amplitude activity during Wake and REM sleep, but have synchronous activity during SWS with periodic high-amplitude waves (Fig 4B).

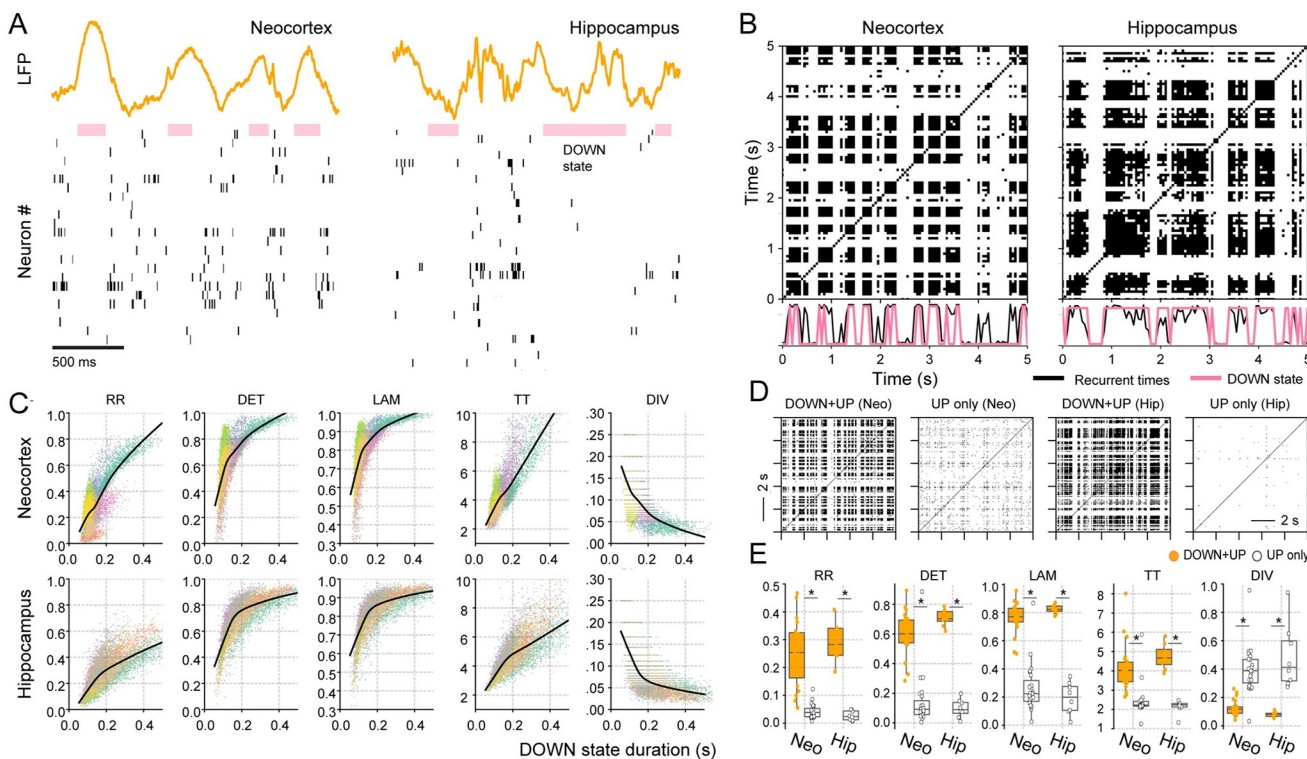

**Fig 3. Correlation between recurrent spiking activity and DOWN states in the neocortex and hippocampus during SWS. A** Example of LFP (different calibrations) and spiking activity in the neocortex (left) and hippocampus (right) exhibiting DOWN states during SWS. **B** Recurrence plots for the corresponding population activity. The number of recurrences per time (sum over columns) is shown in the bottom panel along with the DOWN state periods. **C** Scatter plot between RQA metrics and the average duration of the DOWN state in each recording session, the different colours depict different individual sessions; solid lines indicate the LOWESS regression estimate taking into account all sessions. **D** SWS recurrence plots computed using the whole period (DOWN + UP) or discarding the DOWN states (UP only) for the neocortex (Neo) and hippocampus (Hip). **E** Boxplots of the RQA metrics for DOWN + UP vs. UP only. Neo, $N$ = 24 sessions; Hip, = 9 sessions; $^*p < 0.00001$.

These waves correspond to slow-oscillations of 0–4$Hz$ (Fig 4**C** top), coherent to the LFP activity (Fig 4**C** bottom). Thus, our sLFP recovers the slow-wave activity oscillatory profile present in the real LFP recordings, including a peak in the delta band particularly visible during SWS (compare Fig 4**C** and S5 Fig).

We then quantify the temporal-complexity of LFPs and sLFPs by using Sample Entropy (SE) [40], Permutation Entropy (PE) [41], and Lempel-Ziv Complexity (LZ) [42]. Fig 4**D** shows that results are independent of the chosen complexity measure. The true LFP activity is significantly less entropic during SWS than during REM or Wake (left panels in Fig 4**D**; S4 Table in S1 File), consistent with previous EEG and electrocorticogram (ECoG) results [4–15]. Accordingly, the sLFP exhibits similar temporal-complexity values (right panels of Fig 4**D**; S3 Table in S1 File), and also shows a significant decrease during SWS. Importantly, the complexity reduction during SWS is not easily observed for single units: some neurons decrease while others increase their spiking complexity (S6 Fig), suggesting that the temporal coordination among neurons is necessary for the LFP/sLFP complexity results.

Interestingly, when constructing SWS sLFP only employing UP states (i.e., excluding DOWN states) or actually excluding DOWN state periods from the LFP activity, we find that the decrease in complexity during SWS is lost (Fig 4**D**, S3 Table in S1 File). In fact, the SWS UP states have significantly higher levels of complexity than the SWS sLFP or LFP containing

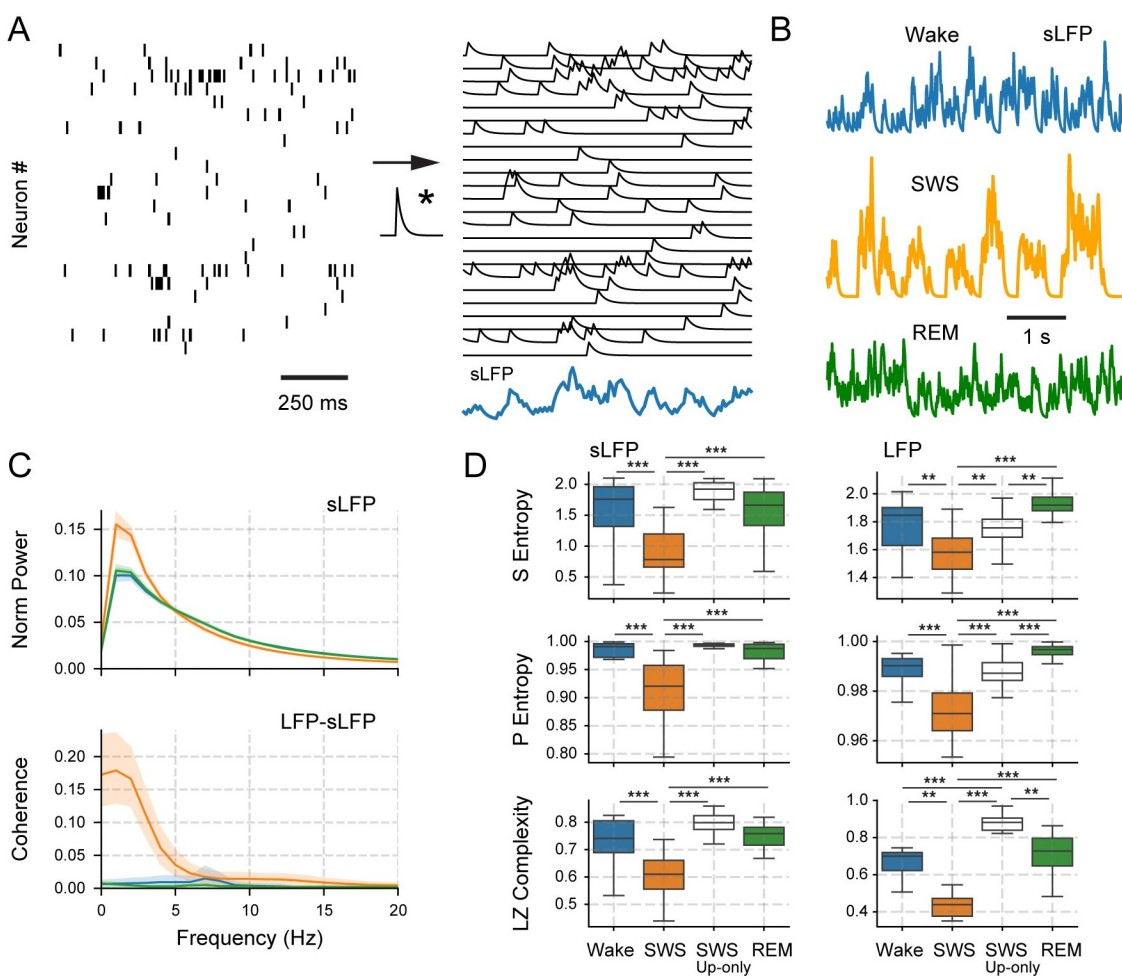

**Fig 4. Construction and analysis of synthetic local fieldpotentials (sLFP) during Wake, SWS, and REM sleep. A** The sLFP is defined as the average of the convolutions between spike trains and a decaying exponential function. **B** Examples of sLFP resulting from Wake, SWS, and REM sleep population activity. **C** sLFP power spectra(top) and coherence between sLFP and LFP (bottom) for the different sleep-wake states (colour coded). **D** Boxplots of Sample Entropy (top), Permutation Entropy (middle), and Lempel-Ziv Complexity (bottom) of the sLFPs and LFPs in each state ($N = 24$ sessions). $^{*}p < 0.05$, $^{**}p < 0.01$, $^{***} = p < 0.001$.

both UP and DOWN states, reaching values comparable to those from Wake or REM states. Therefore, we conclude that DOWN states are necessary for the complexity reduction observed in field recordings since spiking periods are similar across states.

## Spiking periods across states exhibit similar avalanches

Our previous results show that DOWN states disrupt population dynamics in the neocortex and hippocampus (Figs 2–4). We next complement these results by analysing spike avalanches to understand the factors underlying spiking complexity across sleep-wake states. Avalanches are cascades of activity in quiescent systems [43–50], which in our case correspond to active spiking periods within a brain region; by definition, avalanches exclude DOWN states.

Fig 5A shows a neuronal population exhibiting an avalanche, where the time bin defining its occurrence is set as the average inter-spike interval (ISI). By definition, an avalanche starts after a time bin without spikes and finishes when another empty time bin is reached. Two

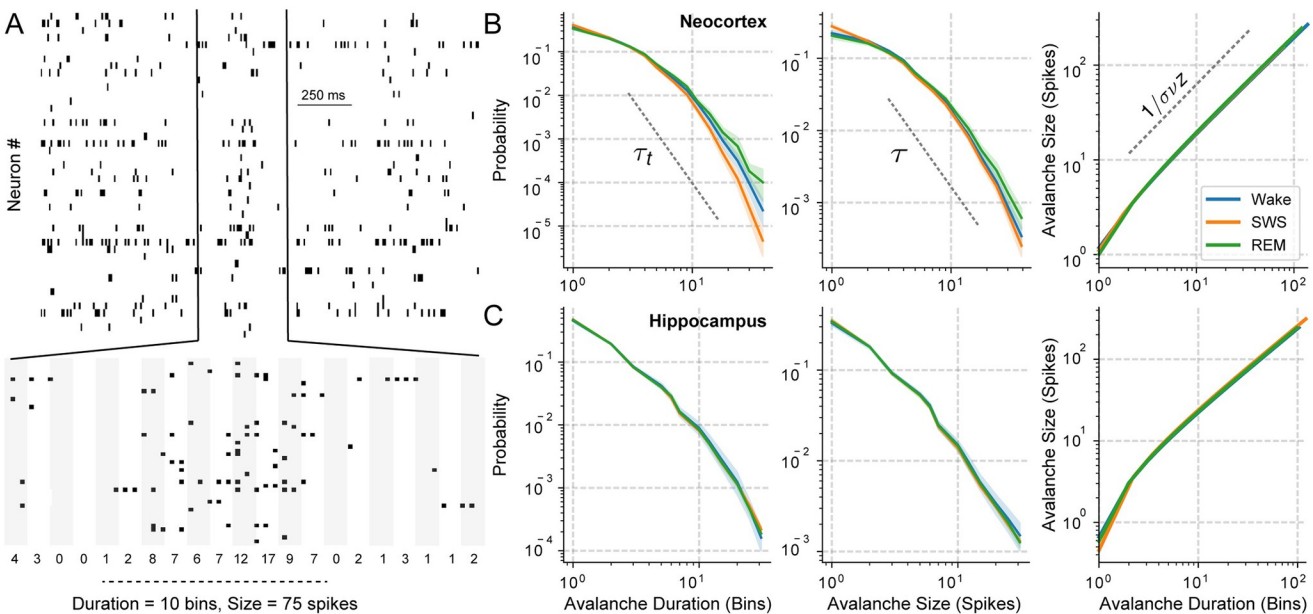

**Fig 5. Avalanche distributions for Wake, SWS and REM sleep. A** Example of a neuronal avalanche. The average ISI is used to bin the raster plot (shaded rectangles) and count the number of spikes per bin. **B** Avalanche statistics for the neocortex. Left: distribution of avalanche duration, used to estimate the $\tau_t$ exponent. Middle: distribution of avalanche size, used to estimate $\tau$. Right: avalanche size as a function of its duration, from which the $\frac{1}{\sigma vz}$ exponent is estimated. **C** As in **B** but for hippocampal avalanches. For each state (colour coded), the mean distributions are shown in solid lines with a shaded area depicting the 95% confidence interval.

parameters commonly characterise an avalanche: its size, i.e., the total number of spikes, and its duration, i.e., the time interval from start to finish. The avalanche statistics for each sleep-wake state are derived from the probability distribution of these parameters [44, 45].

Fig 5B and 5C show minimal differences between the probability distribution of avalanche duration and size during Wake, REM sleep, or SWS in the neocortex and hippocampus, respectively. For instance, avalanches occurring during SWS tend to be shorter due to DOWN states. The power-law exponents for the avalanche duration ($\tau_t$) and size ($\tau$) are related by the crackling noise relationship, $\frac{\tau_t-1}{\tau-1}$, which is a more stringent criticality statistics [45]. Considering all sleep-wake states, we get $\frac{\tau_t-1}{\tau-1} = 1.19$ (inter-quartile range, *IQR* = 0.33) for the neocortex and $\frac{\tau_t-1}{\tau-1} = 1.24$ (*IQR* = 0.37) for the hippocampus, with no significant differences across states ($p$ = 0.31 and 0.68, respectively). More importantly, we find that the avalanche size and duration distributions collapse to the same scaling function resembling a power-law behaviour characterised by the exponent $1/\sigma vz$[45] (right panels in Fig 5B and 5C). This suggests that the spiking periods (UP states) have a common behaviour across sleep-wake states. We find that $\frac{1}{\sigma vz} = 1.11$(*IQR* = 0.05) for the neocortex with no significant differences across sleep-wake states ($p$ = 0.21). Similarly, $\frac{1}{\sigma vz} = 1.20$(*IQR* = 0.02) for the hippocampus ($p$ = 0.09 for state differences).

Thus, once the spiking activity is initiated, it follows a common avalanche regime irrespective of the sleep-wake state. Consequently, complexity differences in the sleep-wake states should originate from DOWN states where no spikes occur. Noteworthy, these results restrict the possible mathematical models which can describe cortical dynamics, since the model must be able to reproduce DOWN states (during SWS) and the avalanches appearing for any state during spiking activity.

## Critical branching model for the spiking activity in the cortex

Here, we show that cortical spiking patterns during wakefulness and sleep can be captured by a critical branching model, known to exhibit universal behaviour [51], when implemented using exponents matching our *in-vivo* results (see Methods). The critical branching model consists of interacting discrete units whose internal state may be resting, spiking, or refractory. The units evolve in time according to the excitation coming from neighbouring units (as controlled by a branching parameter) as well as due to a noisy drive set by a Poisson distribution, which can randomly make a unit fire at any given time. The branching parameter, $\sigma$, determines the probability of a spike from unit A at time $t$ affecting unit B at time $t + 1$. When $\sigma = 1$, the system is critical; the network exhibits a phase transition from a sub- critical quiescent state for $\sigma < 1$ (activity dies out after a small transient) to a super-critical active state for $\sigma > 1$ (activity is self-sustained). The interplay between units interacting due to branching and noise recreates a network of higher-order neurons that receives inputs from lower areas such as the thalamus [52]. To reproduce an SWS state, we add to the branching model a periodic silencing of the noisy drive for some (adjustable percentage of) units in order to model DOWN states.

Fig 6A shows an example of the resultant spike trains for the branching model without (left panel) and with (right panel) the periodic silencing. These results are obtained using 50 units (similar size to the experimental ensembles recorded) and setting the branching parameter at

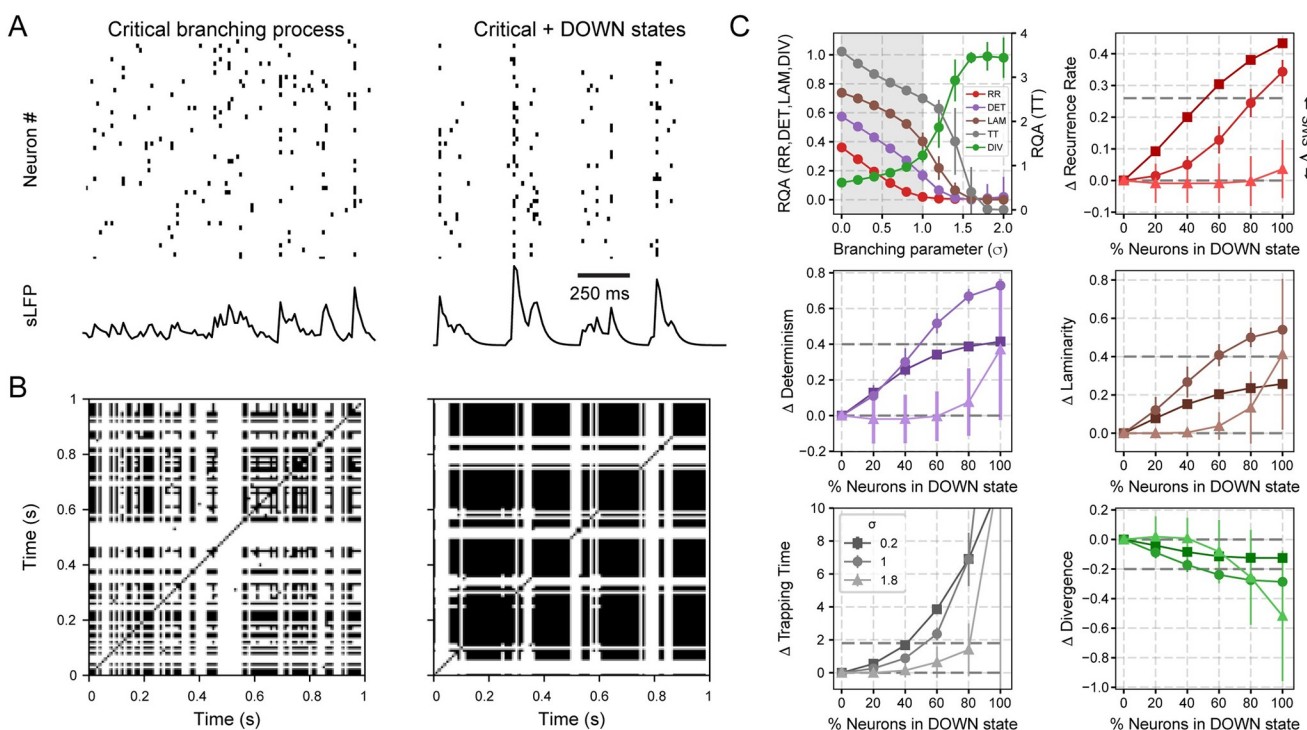

**Fig 6. Critical branching model for neuronal activity during Wake and SWS. A** Left: Population activity (raster plot) and synthetically generated local-field potential (sLFP, as in Fig 4) of a critical branching model with 50 interacting units. The branching parameter is set at $\sigma = 1$ (critical); an excitatory Poisson noise drive each unit independently. Right: DOWN states are generated by periodically silencing (4*Hz*) the noisy drive of a percentage of units. **B** Resultant recurrence plots for the data in **A**. **C** Average (± standard deviation) results from 100 simulations using different network connectivity and initial conditions. Each simulation consisted of $10^6$ iterations in time. Top left: RQA metrics for the original model (i.e., without silencing) as function of $\sigma$; shaded [unshaded] area shows the sub-critical [super-critical] phase. Remaining panels: differences ($\Delta$) between RQA metrics of the original model and the model with periodical silencing as a function of the percentage of neurons having their noise drive silenced (referred to as % of neurons in DOWN state). The horizontal dashed lines show the difference between the actual SWS RQA metrics (Fig 2C) and those of the critical branching model.

the critical point $\sigma = 1$. The respective recurrence plots are shown in Fig 6B. For the modified branching model, we periodically silence the noise input arriving at a given set of units during a 250$ms$ interval (similar to Ref. [44]) and call it Critical + DOWN. This external silencing is enough to synchronize the network to a state of inactivity (Fig 6A), trapping the population trajectory into recurrent square-like patterns (Fig 6B), similar to the experimental results from the neocortex and hippocampus (Fig 2).

We use RQA to quantify the differences between the original and modified branching models. The top left panel of Fig 6C displays RQA metrics for the branching model with 50 units as a function of $\sigma$, where the shaded area marks the sub-critical phase. For $\sigma = 1$, the model has RR $\simeq 0.02$, DET $\simeq 0.2$, LAM $\simeq 0.4$, DIV $\simeq 0.3$, and TT $\simeq 2.5$, which are comparable to the average RQA values of Fig 2C during Wake and REM sleep. The remaining panels show the change in the RQA metrics when the periodic noise-silencing is added to the model—changes are shown as a function of the percentage of units having their noise periodically silenced. The horizontal dashed lines show the difference between the SWS RQA metrics and those of the critical branching model. In other words, this relative SWS metric is found by taking the value obtained from the experimental average RQA metric shown in Fig 2C and subtracting the critical branching model RQA metric from the top left panel in Fig 6C. Using this, we can find the percentage of units with periodically-silenced noise that are needed to reproduce the experimental values found for SWS.

We find that as the number of units with a DOWN state increases (i.e., number of neurons with periodical silencing), the RQA metrics cross those observed during SWS from the *in-vivo* recordings (horizontal dashed line) (Fig 2C). When the model is at the critical point ($\sigma = 1$), a periodic noise-silencing to 40–60% of the units is enough to reproduce the RQA values during SWS (intersection of the $\Delta$ RQA metrics with the corresponding horizontal dashed lines), with the exception of RR, which requires 80%. On the other hand, both the sub- ($\sigma = 0.2$) and super-critical ($\sigma = 1.8$) models need a considerably larger percentage of silencing (80–100%) to reproduce the observed SWS values. Therefore, these results suggest that: i) the branching model needs to be close to $\sigma = 1$ (criticality) to reproduce the recurrent properties observed during Wake or REM sleep, and ii) that the inclusion of a periodic silencing of the noisy drive to 40–60% of the units reproduces the recurrent properties observed during SWS.

## Discussion

Our main findings can be summarised in the following points. The complexity of neuronal dynamics in rats is reduced during SWS owing to spiking patterns repeating more often (i.e., greater recurrences). This spike pattern repetition occurs during DOWN states, thus bridging the decrease in complexity observed in the cellular and field recording levels (such as local field potentials or EEG). Moreover, we reveal a common behaviour in the population spike avalanches appearing across the sleep-wake states (which by definition exclude DOWN states). This scaling makes the sleep-wake states indistinguishable from each other, and demonstrates that the DOWN periods are responsible for the complexity reduction which characterises SWS. Finally, we reproduce these experimental results by numerical experiments employing a critical branching model, suggesting that criticality may favor transitions between states.

### Recurrence quantification analysis improves the study of cortical population dynamics

Our study is based on the analysis of the spiking activity from *in-vivo* population recordings of the neocortex and hippocampus. To that end, we employ RQA, which leads to clear results and interpretations (see Fig 1), is robust to parameter tuning (e.g., changing the tolerance

parameter or time bin width; S1 Fig), and is computationally efficient (e.g., 10$s$ windows are enough to distinguish states). Importantly, RQA allows analysing a population of neurons using various complementing non-linear metrics, such as randomness, entropy, or fractal dimension. This extends the typical characterisation of neuronal dynamics based on single neurons and single metrics (such as the basal firing rate, coefficient of variation, or rhythmicity).

We also compared our results with those provided by topological data analyses (S3 Fig). In particular, persistent homology analyses the topology of a high-dimensional cloud of points (manifold) in phase space [21, 53]. Interestingly, by using this analysis we find that the low-dimensional topology of the neocortical phase-space attractor appears to remain unchanged throughout the sleep-wake cycle (S3 Fig). This contrasts with results from the anterior nucleus of the thalamus which exhibits a ring-like structure during Wake and REM sleep, but not during SWS [53]. Thus, our observations suggest that the dynamical differences across states are still contained within the same manifold.

## Population DOWN states reduce the complexity of cortical activity during SWS

In contrast to the unchanged attractor topology (S3 Fig), our results show that the evolution of neocortical and hippocampal spiking activity is significantly altered during SWS. We show that the cause for this alteration is (mainly) due to the appearance of synchronous DOWN states that disrupt the population spiking patterns and force them into a recurrent, deterministic state. We find a strong correlation between the duration of DOWN states and the number of recurrences in the population activity (Fig 3**B** and 3**C**). Then, we show that the decrease in complexity is lost once we discard the DOWN states from the SWS analysis (Fig 3**D** and 3**E**), making SWS spiking-patterns similar to those from REM sleep or wakefulness.

DOWN states appear to disrupt neuronal patterns in neocortical and hippocampal areas similarly, although both regions have different mechanisms for the generation of DOWN states [25, 35]. During SWS, hippocampal neurons oscillate between long, quiescent, stable periods (without clear membrane hyperpolarization [25]) and bursts of spiking activity (during sharp-wave ripples). In contrast, neocortical neurons oscillate between stable periods of spiking activity and unstable periods of quiescence (associated with hyperpolarization [25]). In spite of these differences, both populations have spiking patterns that are consistent with excitable UP/DOWN states [35].

For individual neocortical neurons, the complexity of firing patterns decreases during SWS [6]. In principle, this decrease could be expected due to the DOWN states, as their appearance causes neurons to remain silent during synchronous intervals. Here, however, when we analyse the firing patterns of individual neurons independently, we find that a considerable number maintain complex patterns even during SWS (S6 Fig). This can occur because either there are DOWN state active neurons, as previously shown in [54], or because the complexity reduction is a collective phenomenon that can only be studied at the population level. We support this latter argument by showing that the difference in complexity between Wake or REM sleep and SWS increases with the number of analyzed neurons (S2 Fig).

## Measuring complexity from synthetic and experimental field recordings

The complex nature of brain recordings—and the decrease in complexity during unconscious states—has been reported using classical neuroscience approaches [55, 56]. For instance, the EEG power spectrum shows a power-law decay, $f^{-\alpha}$, for a broad range of frequencies, referred to as 1/$f$ noise. Interestingly, the exponent $\alpha$ becomes greater than 1 (a more pronounced

decay) during sleep and anaesthesia [55, 56] since DOWN states and slow oscillations promote the appearance of low-frequency power, leading to a steeper spectral decay.

These observations match our sLFP analyses, which recover both the slow oscillations present in true LFPs during sleep (Fig 4C), and their entropy variations during the sleep-wake cycle (Fig 4C)—independently of the chosen entropy metric (i.e., permutation entropy [41], sample entropy [40], and Lempel-Ziv complexity [42]). Notably, the decrease in complexity during SWS is lost when we eliminate the DOWN states from the LFPs and sLFPs (Fig 4E), implying that DOWN states are responsible for reducing the complexity of field recordings during SWS. Consistent with our results, the slow-wave activity (0.1–4Hz) has been associated with the loss of complex neuronal interactions during sleep [32, 57, 58] and is caused by synchronous neuronal DOWN states [24–31]. Of note, the similarities in slow-wave activity [25, 28, 30, 31], and neural complexity [4, 5, 9, 15] between rodents and humans suggest that related mechanisms could also act in the latter during sleep.

It should be noted that estimating neural complexity directly from field recordings might lead to spurious results since there are major differences between the exponent variations in ECoGs and LFPs. For instance, we also find a $f^{-\alpha}$ behaviour in the power spectra of LFPs and ECoGs (S5 Fig) and get similar decay exponents during SWS and REM ($\alpha_{sleep} \simeq 2$). Nevertheless, for ECoGs, we find a significant difference in exponent values from Wake (when $\alpha_{wake} \simeq 1$), while, for LFPs, $\alpha_{sws} \simeq 2$ as during sleep. Thus, this could point to the presence of extra-neural sources during Wake that alter the ECoG power spectrum decay but do not influence the LFP recording level. Therefore, we argue that complementing field recordings with spiking activity is necessary to unveil and study genuine neural complexity.

## Spiking periods show similar dynamics across states

An important result verified through complementing approaches (Figs 3–5) is that while spiking activity is occurring, SWS behaves similar to Wake or REM sleep. Thus, we suggest near-critical dynamics might be a necessary (but not sufficient) condition for neural complexity. We show that neuronal avalanches of length $t$ contain an average of $g(t)$ spikes, where $g$ is a scaling function independent of the sleep-wake state. This means that avalanches from the frontal cortex and hippocampus of rats across states follow a close behaviour that resembles a power law (Fig 5B and 5C), similar to previous results in the visual cortex [44]. We find that the exponents $\frac{1}{\sigma vz}$ and $\frac{\tau_t - 1}{\tau - 1}$ are relatively close in both areas (which follows the crackling noise relationship, claimed as a more stringent criticality test [45]). Specifically, for the neocortex, we have $\frac{1}{\sigma vz} = 1.11$ and $\frac{\tau_t - 1}{\tau - 1} = 1.19$, and for the hippocampus, we have $\frac{1}{\sigma vz} = 1.20$ and $\frac{\tau_t - 1}{\tau - 1} = 1.24$. These similar exponents are expected if the system is close to a critical point and have been reported for intermediate levels of spiking variability in anaesthetized rats and freely behaving mice [45]. Therefore, our results support the hypothesis that complex cortical activity arises from near-critical dynamics [43–47, 49, 50, 59–62].

## DOWN states are sufficient to reproduce the complexity reduction in a critical model of the cortex

To complement our *in-vivo* results, we show that introducing DOWN states into a critical branching model is sufficient to generate an SWS-like state (Fig 6A and 6B). We achieve this by periodically silencing the noisy drive to a given percentage of units, thus mimicking the synaptic input reduction to pyramidal cells during SWS in the neocortex [63]. This reduction is likely caused by a pre-synaptic GABAb inhibition of the excitatory inputs arriving at the apical dendrites of principal cells [64], coordinated by the thalamus [65]. In contrast to neocortical

mechanisms, UP/DOWN states in the hippocampus are related to sharp-wave ripple generation, where low-spiking DOWN states predominate, and UP-states are initiated by recurrent excitation from CA3 neurons [66]. Therefore, in the hippocampus, the periodic silencing reproduces DOWN states occurring between sharp-wave ripples.

In our model, there is no need to silence the input to 100% of the neurons to reproduce the experimental results, consistent with the lack of hippocampal OFF-periods and minimal firing levels during DOWN states [35]. Additionally, we note that a similar strategy has been employed to model slow-wave oscillations during anaesthesia [44]. Moreover, we find that being near the critical point (Fig 6C) allows for more flexible transitions to the SWS-like state with respect to the sub- or super-critical model. Specifically, silencing the input to 40–60% creates a decrease in complexity similar to that observed experimentally. Notice further that, despite the subcritical model requiring less silenced neurons to achieve RR levels, it fails to capture LAM and DIV SWS values. These results further add to the idea of criticality in the brain, which would explain increased complexity [67], information processing and transmission [43], and dynamical range [60].

## Conclusion

Complexity has been suggested as a necessary condition for cognition [14, 68]. Accordingly, it has been widely reported that during SWS the complexity of brain dynamics decreases [4–14]. However, the reason why brain signals are complex when animals are awake or why this complexity is lost during unconscious remains controversial [69]. In the present work, we conclude that DOWN states fully account for the complexity decrease during SWS, while a common underlying spiking regime describes all sleep-wake states in the neocortex and hippocampus.

## Materials and methods

### Datasets

Datasets We analyse 2 datasets: Watson et al. (neocortex, available at CRCNS.org/fcx) [70]; and Grosmark and Buzsaki (hippocampus, available at CRCNS.org/hc) [71]. The reader is referred to the original publications for details about experimental methods. We provide a summary below.

For the neocortex dataset [70], silicon probes were implanted in frontal cortical areas of 11 male Long Evans rats. Recording sites included medial prefrontal cortex (mPFC), anterior cingulate cortex (ACC), pre-motor cortex/M2, and orbitofrontal cortex (OFC). Recordings took place during light hours in the home cage (25 sessions, mean ± SD duration of 4.8 ± 2.2 h). We note that we exclude *BWRat19_032413* from the analysis since it did not contain REM sleep. Data was sampled at 20 *kHz*. To extract LFPs, recordings were low-pass filtered and re-sampled at 1.25 *kHz*. To extract spikes, data was high-pass filtered at 800 *Hz*, and then threshold crossings were detected. Spike sorting was accomplished by means of the Klusta-Kwik software. Sleep-wake states were identified by means electrophysiological and EMG analyses [70]. OFF periods were extracted as periods of population silence lasting at least 50*ms* and no more than 1250*ms*. Conversely, ON periods consisted of periods of population firing between OFF periods with at least 10 total spikes and lasting 200–4000*ms*.

For the hippocampus dataset [72], 7 silicon probes were implanted in the dorsal CA1 of 4 male Long Evans rats. LFP and spikes were extracted the same way as in the neocortex dataset; similar criteria were employed to identify the sleep-wake states. DOWN [UP] states were identified during SWS selecting the times when less [more] than 10% of neurons fired.

## Recurrence quantification analysis

Prior to analyse the recurrences of the neuronal population [73–75], we bin the spike train of each neuron using 50*ms* non-overlapping spike count windows. The dynamics of the neuronal population is then described by the evolving firing counts of all neurons, which defines a trajectory (with a time resolution of 50*ms*) in the population phase space (that has *N* dimensions for *N* neurons).

A recurrence plot of the evolving firing counts is defined by the symmetric matrix

$$
\begin{cases}
R(i,j) = 1, & \text{if } \|\vec{x}(t_i) - \vec{x}(t_j)\| \leq \epsilon, \\
R(i,j) = 0, & \text{otherwise,}
\end{cases}
\tag{1}
$$

where $\vec{x}(t_i)$ $[\vec{x}(t_j)]$ is the phase-space vector containing the firing counts of all neurons at the time bin $t_i$ $[t_j]$, with $i = 1, \ldots, T$ (*T* being the number of 50*ms* bins that are available from the spike-train signals, e.g., *T* = 200 when using 10*s* windows) and $\epsilon > 0$ is the tolerance parameter defining closeness. We set $\epsilon = \sigma_P$, where $\sigma_P$ is the standard deviation (across time) of the summed firing counts (across neurons) during wakefulness. $R(i,j) = 1$ corresponds to having the trajectory of the neuronal population at time $t_i$ returning to the same region (up to $\epsilon$) of phase space that it was at time $t_j$; that is, a recurrence happens after $t_i - t_j$.

To quantify the patterns arising from recurrences, we employ common measures from Recurrence Quantification Analysis (RQA) [73–75]. The metrics we use are: recurrence rate (RR), determinism (DET), laminarity (LAM), trapping time (TT) and divergence (DIV), which are defined by

$$
RR = \frac{1}{N^2}\sum_{i,j=1}^{N} R_{i,j}, \quad DET = \frac{\sum_{l=l_{\min}}^{N} lP(l)}{\sum_{l=1}^{N} lP(l)}, \quad LAM = \frac{\sum_{v=v_{\min}}^{N} vP(v)}{\sum_{v=1}^{N} vP(v)},
$$

$$
TT = \frac{\sum_{v=v_{\min}}^{N} vP(v)}{\sum_{v=v_{\min}}^{N} P(v)}, \quad DIV = \frac{1}{Lmax},
$$

where $P(l)[P(v)]$ indicates the probability of finding a diagonal [vertical] line of length $l[v]$, and *Lmax* indicates the longest diagonal line excluding the identity line.

## Synthetic LFPs and field complexity measures

We construct synthetic Local Field Potentials (sLFPs) by averaging the convolutions between spike counts in 80*ms* non-overlapping bins of each excitatory neuron and an exponentially decreasing kernel. Namely, $C_n(t) = S_n(t) \star \exp(-t/\tau)$, where $S_n(t)$ is the *n*-th neuron spike count time series, $\tau = 24ms$ is the characteristic time-scale of the kernel (typical mEPSP time for pyramidal neurons in the frontal cortex [76]), and $\star$ the convolution operator.

The resultant sLFP is then obtained from

$$
sLFP(t) = \frac{1}{N}\sum_{n=1}^{N} C_n(t),
\tag{2}
$$

where *N* is the number of simultaneously recorded neurons.

For the frequency analysis, we compute the power spectrum of the sLFP using Welch's algorithm. We apply the `signal.welch` scipy python 3 function (scipy.org), with a 1*s* moving Hanning window (without overlap), and a 1*Hz* frequency resolution. For computing the sLFP-LFP coherence, we first downsample the LFP recordings to 125*Hz* and average them

across channels; we then employ the `signal.coherence` scipy function, using the same parameters as the power spectrum. We note that the 80$ms$ spike count bin equals a 125 Hz sampling frequency.

For measuring sLFP and LFP complexity, we use Permutation Entropy (PE) [41], Sample Entropy (SE) [40], and Lempel-Ziv (LZ) [42] Complexity, implemented through the `antroPy` python 3 package (github/antropy). Prior to computing these measures, we also downsample the LFP recordings to 125$Hz$ and average them across channels.

PE [41] requires dividing the sLFP or the average LFP signal, $\{x(t), t = 1, \ldots, T\}$, into $\lfloor (T - D)/D \rfloor$ non-overlapping vectors of $D$ data points, with $D \ll T$ (shorter than the time-series length). Then, each vector is classified as a symbol $\alpha$ according to the number of permutations needed to order its $D$ elements. We employ $\tau = 5$, where $\tau$ is the distance between consecutive time-stamps inside each vector containing $D = 3$ time points. Finally, the PE [41] is the Shannon entropy [77] of the resultant symbolic sequence; that is, $H = -\Sigma_\alpha p(\alpha) \log [p(\alpha)]$, where $p(\alpha)$ is the probability of finding symbol $\alpha$ in the signal.

Similar to PE, SE [40] consists of dividing a time-series into a series of $D$-sized vectors ($\vec{y}_D(i) = \{x(t_i), x(t_{i+1}), \ldots, x(t_{i+D-1})\}$) and is defined as $SE = -log\left(\frac{A}{B}\right)$, where $A$ and $B$ are, respectively, the number of times that $d[\vec{y}_{D+1}(i), \vec{y}_{D+1}(j)] < r$ and $d[\vec{y}_D(i), \vec{y}_D(j)] < r$ for all $i, j$ vector pairs, and $d$ is the Chebyshev distance and $r > 0$ is a tolerance parameter (0.1 * $SD$ of the signal). In our case $D = 3$, and we downsample the signals by a factor of 5 in order to match $\tau$ from PE.

LZ [42] complexity is estimated by the `LZ-76` algorithm. We start by creating a binary sequence from the mean value of the sLFP or the average LFP recording—all points larger than the signal mean are converted to 1, and 0 otherwise. Then, we count the number of different binary sub-strings from beginning to end, #*substrings*. The LZ complexity is given by $LZ_w$ = (#*substrings*)/($w/ \log(w)$), where $w$ is the length of the binary sequence.

## Neuronal avalanches

We quantify neuronal avalanches following previous studies [44, 45]. First, population activity is binned employing the average inter-spike interval. Then, we measure the time (duration) and number (size) of spikes between one empty bin (0 spikes) to the following empty bin. We use the `powerlaw` (pypi.org/powerlaw) python 3 package to construct the probability distributions and obtain their exponents: $\tau_t$ and $\tau$. We also compute the average number of spikes as a function of the avalanche duration, and obtain the exponent $\frac{1}{v\sigma z}$ by means of an ordinary least square fit on the log-log scale distribution.

## Critical branching model

The critical branching model consists of 50 interacting units randomly connected in an Erdös-Rényi topology with a 0.03 attachment probability (i.e., each pair of units has a 0.03 probability of having a link). The time step was set as 1$ms$. Each unit has 3 possible states: resting, firing or refractory. The transition between resting and firing can either occur from the excitation coming from a connected neuron firing in the preceding time, or by the intrinsic Poisson noise that each neuron receives independently. The excitatory Poisson noisy drive is set by generating a random matrix whose values come from a [0, 1] uniform distribution, and then setting for each entry a spike if the value is less than $1 - e^{-\lambda}$ ($\lambda = 0.014$). We periodically silenced the Poisson noise for 250$ms$ at a 4$Hz$ frequency to create a SWS-like state. Once a neuron fires, it goes to the refractory state and it cannot be excited again. After one step in the refractory state, the neuron goes to the resting state and becomes excitable again. The propagation of spikes is controlled by the branching parameter $\sigma$, which regulates the overall excitability of the system.

For instance, if neuron *i* fires, the probability that a connected unit fires is defined as $P_{prop} = \sigma/\langle k_j \rangle$, where $\langle k_j \rangle$ is the average node degree across all units *j*.

## Statistics

We present data as boxplots showing the median, the 1st and 3rd quartiles, and the whiskers corresponding to 1.5 times the inter-quartile range. Because of the non-Gaussian distributions of the complexity metrics, we employ non-parametric statistics. We use the Friedman test available with the `scipy.stats` from python 3 package to compare the results among states, i.e., Wake-SWS-REM (Wake-SWS-SWSup-REM, Fig 4**D**), with the Siegel post-hoc test applying the Benjamini-Hochberg false discovery rate correction available with the `scikitlearn` (scikit-learn.org). We set $p < 0.05$ for a result to be considered significant. In addition to p-values, we also report Cohen's d, which quantifies the magnitude of a result in terms of a standardised difference between conditions; an effect size is considered to be large if Cohen's *d* is $> 0.8$. For the power spectra and avalanche results, we present the data as the mean with the 95% confidence interval (obtained through bootstrap sampling). For the correlation analysis, we employ LOWESS regression to fit the best estimate to the scatter plot by means of the `regplot` python 3 function available at seaborn.pydata.org. As LOWESS regression has no associated *p* value, we employ a linear regression for each session and consider the result as significant only if $p < 0.05$ for all sessions. Additionally, to correlate the DOWN states to the recurrence sum, we employ the point-biserial correlation `pointbiserialr` function available at scipy.org.

## Supporting information

**S1 Fig. RQA differences among states are robust to parameter choice. A** RQA metrics for different tolerance levels $\epsilon$ defining recurrence in phase space. We vary $\epsilon$ from 0 std to 4 std of the population firing counts. Setting $\epsilon$ to 0 means that a recurrence occurs between two times for the exact same neuronal firing pattern. The time bin is kept fix at 50 ms. **B** RQA metrics for different time binning of the population activity. Time bins are changed from 20 ms to 200 ms in order to define the firing counts for each neuron. The $\epsilon$ is kept fix at 1 std. The mean and its corresponding 95% confidence intervals are shown for each plot.
(TIF)

**S2 Fig. RQA differences between states correlate with the number of neurons recorded.** Absolute RQA differences between states as a function of the number of simultaneously recorded neurons. Each dot shows a recording session while the solid line the linear regression estimate with its 95% confidence interval. **A** shows the SWS-Wake difference, while **B** the SWS-REM difference.
(TIF)

**S3 Fig. Persistent Homology cannot distinguish the sleep-wake states in the neocortex.** Top panels: Point clouds obtained after dimensionality reduction. A representative animal is shown during Wake, SWS and REM sleep. Bottom panels: Betti 0 (HO) and Betti 1 (H1) barcodes for the same animal shown in the top panel. The length of each bar shows the level of persistence of each Betti 0 and 1 component.
(TIF)

**S4 Fig. UP state recurrences are similar to Wake or REM sleep. A** Recurrence plots constructed from a 10*s* interval of the population activity using. **B** 5 RQA metrics for the sleep-wake states; boxplots show results from the pool of 24 sessions across 12 animals (outliers are

not shown).
(TIF)

**S5 Fig. Power spectrum slope differs among states. A** LFP [ECoG] recordings coming from the frontal cortex [M1 cortex] during the states of Wake, SWS and REM sleep. The mean and its corresponding 95% confidence intervals are shown for each plot. **B** Power spectrum exponents calculated through ordinary least-squares fit on a log-log scale (OLS) or through the FOOOF parametrized spectra (FOOOF) [78] which only includes the aperiodic component.
(TIF)

**S6 Fig. Single neurons deviate from the ensemble behaviour.** Lempel-Ziv Complexity of single neuron firing pattern between Wake and SWS. Each bar shows the total number of neurons or sessions whose temporal complexity decreased or increased during sleep. Left: LFP recordings. Middle: sLFP recordings- Right: Single unit recordings.
(TIF)

**S1 Text. Supplementary methods [9, 21, 53].**
(PDF)

**S1 File.**
(ZIP)

## Author Contributions

**Conceptualization:** Joaquín González, Adriano B. L. Tort, Pablo Torterolo, Nicolás Rubido.

**Formal analysis:** Joaquín González, Nicolás Rubido.

**Funding acquisition:** Joaquín González, Adriano B. L. Tort, Pablo Torterolo.

**Investigation:** Joaquín González.

**Methodology:** Joaquín González, Matias Cavelli.

**Resources:** Adriano B. L. Tort.

**Software:** Joaquín González.

**Supervision:** Adriano B. L. Tort, Pablo Torterolo, Nicolás Rubido.

**Visualization:** Joaquín González.

**Writing – original draft:** Joaquín González, Nicolás Rubido.

**Writing – review & editing:** Joaquín González, Matias Cavelli, Adriano B. L. Tort, Pablo Torterolo, Nicolás Rubido.

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
