## [Decision Letter · Decision Letter 0]

6 Jun 2023

PONE-D-23-12688Sleep disrupts complex spiking dynamics in the neocortex and hippocampusPLOS ONE

Dear Dr. Gonzalez,

Thank you for submitting your manuscript to PLOS ONE. After careful consideration, we feel that it has merit but does not fully meet PLOS ONE’s publication criteria as it currently stands. Therefore, we invite you to submit a revised version of the manuscript that addresses the points raised during the review process.

We look forward to receiving your revised manuscript.

Kind regards,

Jordi Garcia-Ojalvo

Academic Editor

PLOS ONE

“J.G acknowledges the support of Comisión Académica de Posgrado (CAP),

CSIC Iniciación and PEDECIBA. P.T also acknowledges the support of PEDECIBA.

A.B.L.T acknowledges the support of CAPES and CNPq. N.R. acknowledges

the CSIC group grant “CSIC2018 - FID 13 - Grupo ID 722”

“J.G acknowledges the support of Comisi´on Acad´emica de Posgrado (CAP), CSIC Iniciaci´on and PEDECIBA. P.T also acknowledges the support of PEDECIBA. A.B.L.T acknowledges the support of CAPES and CNPq. N.R. acknowledges the CSIC group grant “CSIC2018 - FID 13 - Grupo ID 722”

“J.G acknowledges the support of Comisión Académica de Posgrado (CAP),

CSIC Iniciación and PEDECIBA. P.T also acknowledges the support of PEDECIBA.

A.B.L.T acknowledges the support of CAPES and CNPq. N.R. acknowledges

the CSIC group grant “CSIC2018 - FID 13 - Grupo ID 722”

4. We noted in your submission details that a portion of your manuscript may have been presented or published elsewhere. “Yes, We analyse 2 publicly available datasets: Watson et al. (https://crcns.org/data-sets/fcx/fcx-1); and Grosmark and Buzsaki (https://crcns.org/data-sets/hc/hc-11)” Please clarify whether this publication was peer-reviewed and formally published. If this work was previously peer-reviewed and published, in the cover letter please provide the reason that this work does not constitute dual publication and should be included in the current manuscript.

Reviewers' comments:

Reviewer's Responses to Questions

**Comments to the Author**

1. Is the manuscript technically sound, and do the data support the conclusions?

Reviewer #1: Yes

Reviewer #2: Partly

2. Has the statistical analysis been performed appropriately and rigorously? 

Reviewer #1: Yes

Reviewer #2: Yes

3. Have the authors made all data underlying the findings in their manuscript fully available?

Reviewer #1: Yes

Reviewer #2: No

4. Is the manuscript presented in an intelligible fashion and written in standard English?

Reviewer #1: Yes

Reviewer #2: Yes

5. Review Comments to the Author

Reviewer #1: Below I list my comments, I hope they will help to improve the manuscript. Please consider comments 4, 12, 13 most important.

1. I would specify already in the abstract, author summary, discussion, etc. what data is the study conducted on, in particular on what species.

2. Similarly, many references and phenomena cited by the authors refer to humans, whereas the study is performed on an open dataset collected on rodents. Therefore, I'd find it useful to discuss how are (could be) the published results relevant to humans.

3. In my view the subsection 'Recurrence analysis reduces high-dimensional dynamics to a 2D representation' is rather Methods than Results.

4. A general comment about figures: I find them too small. I am not sure if they will be in the same size in the final version of the manuscript, but in the version for revision I had sometimes problems to read details. See for example the rat brain in top-left corner of figure 2, it is completely not readable.

5. Figure 2: I would suggest showing only a projection of the 5 dimensional space, e.g. only n1, n2 and n3. Currently it is not very clear. Also, maybe make clear that only the first matrix from panel C corresponds to panels A and B.

6. “DIV quantifies the chaoticity of the trajectory” – is it strict? Chaos is characterized with an exponential divergence, perhaps here one could observe a large divergence that would not be exponential and thus not chaotic. Please discuss or change the wording.

7. In the definition of DIV you could add that the matrix diagonal is not taken into account (I understand it would always be present and have length N).

8. I am not convinced that DET measures smoothness of trajectories, imagine a relaxation oscillator which produces abruptly changing trajectories, although is deterministic.

9. “In fact, when comparing the RQA metrics among the secondary motor cortex (M2), medial prefrontal cortex (mPFC), orbitofrontal cortex (OFC), anterior cingulate cortex (ACC) and the CA1 hippocampus region, we find no statistical difference” - what statistical test was used? Also, I more often see p-value written with a lowercase letter.

10. Figure 2: I spot three typos: a Spanish accent over the “o” in “cortex”, a space after the hyphen in “orbito-frontal” and no space after the “E” panel symbol.

11. The last paragraph of the section “Neuronal recurrences during SWS are mainly driven by DOWN states” is written twice in two different versions.

12. Honestly, I am confused about the LFP / sLFP. I understand that LFP is in the dataset. Please state it clearly already when you introduce the results related to LFP. Why would you need to use sLFP at all? If you have a real LFP you could crop it and analyze UP/DOWN periods separately, similarly as you do with sLFP. If you don’t have the real LFP, what do you present in the figure then? Could you clarify it?

13. (In the context of Figure 4) One of the messages of the paper is that discarding DOWN states leads to dynamics resembling REM / Wake. In order to prove this thesis one would need to perform a statistical test on two samples: one from only UP states and the other from REM and/or Wake, whereas I only find a test comparing DOWN+UP with only UP. From this test solely one can not draw the above mentioned conclusions.

14. There is a space bar missing at “Accordingly, the sLFP…”.

15. Figure 5: is the sigma-nu-z index well written? I am not familiar with it, but it seems a bit puzzling that three symbols are used for one quantity. Perhaps some lower/upper indexing is lost?

16. Avalanches discussion: I would be curious if a DOWN state can occur during an avalanche, thus cutting it. On one hand it is claimed in the manuscript that the avalanches are the same during UP and REM / Wake, but then in figure 5 B we see that for SWS avalanches tend to be smaller and shorter (the orange line is the lowest) which perhaps could be explained by the cutting effect.

17. “suggesting that criticality allows for optimal transitions between states.” - how is this claim supported by the results? They only show that avalanches are present and that the model set close to criticality can reproduce them. But transitions between UP/DOWN are imposed onto the model by silencing some stochastic inputs, thus I do not see how are these transitions facilitated by criticality.

18. In methods silicon probes are mentioned, they record LFP. But not EcoG stripes are mentioned, how were these data collected then?

19. I think tables S3 and S4 should be referring to figures 4, not 3.

Reviewer #2: In this paper, the authors analyze in-vivo recordings from frontal cortical areas and from dorsal CA1 of rats, containing neuronal activity during wakefulness, slow-wave sleep (SWS) and rapid-eye-movement sleep (REM). The authors characterize the dynamics in the phase-space of neuron’s activity by quantifying the amount of recurrence. The authors found that SWS have less complex dynamics compared to awake and REM states, and that this difference can be explained by the presence of DOWN states. Next, they complete this study with analysis on local field potentials (LFPs) and spike avalanches. Finally, they propose a critical branching model to explain their results. The results are interesting and the structure of the manuscript is overall clear. An important point of the manuscript is the link between the level of complexity observed at the macroscopic level, and the differences in the neuronal activity during each state. However, I find that some of the main findings are poorly described, and parts of the text are difficult to understand. I describe here my points of concern.

Major comments:

(1) The results in Figure 1 are poorly described. The authors used several different measures of complexity, and I believe the authors should improve the description of these measures, and don’t list them twice in the text. Together with this, I suggest the authors should improve the description of the possible recurrence plots in Panel C, and justify why these artificial examples are important to understand the following results.

(2) In general, the text in the results section can be improved. For example, in page 5: the last two paragraph are repeated, with few words being replaced. The authors should decide which version to keep. Another example, in page 3: the sentence “An attractor is evidenced as a manifold that attracts different trajectories of the system to the same region of the phase-space; the more convoluted (fractal) the attractor is, the higher the temporal complexity of its trajectories” and the following sentence should be moved somewhere else. If it is there to justify the use of RQA, then I believe there is no need for it.

(3) In Figure 4, I don’t understand the need to use surrogate LFP instead of just using the LFP from the data. The authors first justify the use of sLFP as “The motivation behind this method is that it allows to precisely control the sources which dictate the field potential and avoid the influence of any external variable not directly related to spiking activity”. Then, they show in panel C how the sLFP is similar to the LFP, and in panel D that the entropy methods they used lead to very similar results when applied to either LFP or sLFP. This, to me, suggest that there is no real difference between the two types of signals in terms of complexity. However, the results in panel E support the main claim of the paper, but they are shown only for the sLFP. It is important to see that these results hold true also for the LFP data. I suggest the authors to apply the analysis of panel E on the raw LFP, to move the results on the sLFP in the supplementary, and to keep the results on the raw data LFP as the main point. Otherwise, give an explanation on why the raw LFP cannot be analysed as in panel E.

(4) The authors referred to some supplementary figures (Figure S4, S5 and S6) only in the Discussion section. I believe the authors should refer to those figures in the Result section.

Minor comments:

(1) I find that all the figures of the paper are too small compared to the text. One needs to pretty much zoom in to be able to see and understand the results. I am not aware if this problem can be solved during formatting. I suggest that the authors update the figures to increase their readability.

(2) I believe that the average reader is not necessarily familiar with the concept of “complexity”. I suggest the author to better define the notion of the complexity of a neuronal signal in the text, especially the abstract and the introduction.

(3) Page 5: “The square-shaped recurrences appearing during SWS in Fig.2 can be generated by two possible mechanisms. Either the neurons remain constantly active for a period of time, or they remain silent (null firing counts) and give rise to a trajectory in the origin of the phase space. Next, we show that the latter is true and is mainly due to DOWN states“. I do not understand why a squared-shaped recurrence could not be given by some neuron firing and some remaining silent, as along as they keep this activity for a period of time.

(4) Figure 6: I suggest the authors should show also some metrics of variation around the mean, and consider doing a statistical analysis to quantify the significance of the difference in the models.

6. PLOS authors have the option to publish the peer review history of their article (what does this mean?). If published, this will include your full peer review and any attached files.

Reviewer #1: **Yes: **Maciej Jedynak

Reviewer #2: **Yes: **Matteo Saponati

---

## [Author Response · Author response to Decision Letter 0]

10 Jul 2023

Response to the editor:

We thank the editor for the work done handling our manuscript. We have now employed the PLOS latex template and made all amendments necessary to comply with the additional requirements.

“J.G acknowledges the support of Comisión Académica de Posgrado (CAP),

CSIC Iniciación and PEDECIBA. P.T also acknowledges the support of PEDECIBA.

A.B.L.T acknowledges the support of CAPES and CNPq. N.R. acknowledges

the CSIC group grant “CSIC2018 - FID 13 - Grupo ID 722”

“J.G acknowledges the support of Comisi´on Acad´emica de Posgrado (CAP), CSIC Iniciaci´on and PEDECIBA. P.T also acknowledges the support of PEDECIBA. A.B.L.T acknowledges the support of CAPES and CNPq. N.R. acknowledges the CSIC group grant “CSIC2018 - FID 13 - Grupo ID 722”

“J.G acknowledges the support of Comisión Académica de Posgrado (CAP),

CSIC Iniciación and PEDECIBA. P.T also acknowledges the support of PEDECIBA.

A.B.L.T acknowledges the support of CAPES and CNPq. N.R. acknowledges

the CSIC group grant “CSIC2018 - FID 13 - Grupo ID 722”

We have removed those statements from the Acknowledgments Section.

4. We noted in your submission details that a portion of your manuscript may have been presented or published elsewhere. “Yes, We analyse 2 publicly available datasets: Watson et al. (https://crcns.org/data-sets/fcx/fcx-1); and Grosmark and Buzsaki (https://crcns.org/data-sets/hc/hc-11)” Please clarify whether this publication was peer-reviewed and formally published. If this work was previously peer-reviewed and published, in the cover letter please provide the reason that this work does not constitute dual publication and should be included in the current manuscript.

We note that both datasets come from two papers already published and cited in our work: 

Watson, B. O., Levenstein, D., Greene, J. P., Gelinas, J. N., & Buzsáki, G. (2016). Network Homeostasis and State Dynamics of Neocortical Sleep. Neuron, 90(4), 839–852. https://doi.org/10.1016/j.neuron.2016.03.036

Grosmark, A. D., & Buzsáki, G. (2016). Diversity in neural firing dynamics supports both rigid and learned hippocampal sequences. Science (New York, N.Y.), 351(6280), 1440–1443. https://doi.org/10.1126/science.aad1935

Response to the reviewers:

Reviewer #1: Below I list my comments, I hope they will help to improve the manuscript. Please consider comments 4, 12, 13 most important.

1. I would specify already in the abstract, author summary, discussion, etc. what data is the study conducted on, in particular on what species.

First of all, we would like to thank the reviewer for his work. We now inform the species employed in both the Abstract, Author Summary, and Discussion. 

2. Similarly, many references and phenomena cited by the authors refer to humans, whereas the study is performed on an open dataset collected on rodents. Therefore, I'd find it useful to discuss how are (could be) the published results relevant to humans.

We thank the reviewer for his comment. We now include the following sentence in the Discussion: “Of note, the similarities in slow-wave activity [25, 28, 30, 31] and 

neural complexity [4, 5, 9, 15] between rodents and humans suggest that related 

mechanisms could also act in the latter during sleep”.

3. In my view the subsection 'Recurrence analysis reduces high-dimensional dynamics to a 2D representation' is rather Methods than Results.

We thank the reviewer for his comment. We feel, however, that such a description is needed in the beginning of the Results section in order to clarify the method and make the average reader understand the subsequent findings. We also note that, upon a request from Reviewer 2, we have edited this section (and associated Figure 1) to let it more palatable. 

4. A general comment about figures: I find them too small. I am not sure if they will be in the same size in the final version of the manuscript, but in the version for revision I had sometimes problems to read details. See for example the rat brain in top-left corner of figure 2, it is completely not readable.

We thank the reviewer for this comment, which we agree. We believe that the figures will be larger in the final PDF version of our manuscript (so as to extend beyond the text margins). In any case, we have increased the font size in several of the figures.

5. Figure 2: I would suggest showing only a projection of the 5 dimensional space, e.g. only n1, n2 and n3. Currently it is not very clear. Also, maybe make clear that only the first matrix from panel C corresponds to panels A and B.

We thank the reviewer for his comment. We have now updated Figure 1 (pasted below) according to the comments made, i.e., only showing a 3 neuron example and improving the depiction of both how phase-space activity translates into the recurrence plot, and also showing how dynamics of the three model examples are captured by the RQA metrics. 

6. “DIV quantifies the chaoticity of the trajectory” – is it strict? Chaos is characterized with an exponential divergence, perhaps here one could observe a large divergence that would not be exponential and thus not chaotic. Please discuss or change the wording.

We agree with the logic of the comment and have thus changed the wording (at this and other text passages).

7. In the definition of DIV you could add that the matrix diagonal is not taken into account (I understand it would always be present and have length N).

We thank the reviewer for the comment. We have now added that information in the Results section: “Divergence (inverse of the longest diagonal line, excluding the identity line)”. 

8. I am not convinced that DET measures smoothness of trajectories, imagine a relaxation oscillator which produces abruptly changing trajectories, although is deterministic.

To avoid confusions or misconceptions, we have made major revisions in that entire paragraph to now show how each RQA metric behaves for each example of Figure 1.

9. “In fact, when comparing the RQA metrics among the secondary motor cortex (M2), medial prefrontal cortex (mPFC), orbitofrontal cortex (OFC), anterior cingulate cortex (ACC) and the CA1 hippocampus region, we find no statistical difference” - what statistical test was used? Also, I more often see p-value written with a lowercase letter.

We have now clarified that a Kruskal-Wallis test was performed across cortical locations. Of note, there is also no cortical differences when using Two-Way ANOVA (state, cortex). We also changed all p-value instances to lower case.

10. Figure 2: I spot three typos: a Spanish accent over the “o” in “cortex”, a space after the hyphen in “orbito-frontal” and no space after the “E” panel symbol.

We thank the reviewer for spotting such errors, which we have now fixed. 

11. The last paragraph of the section “Neuronal recurrences during SWS are mainly driven by DOWN states” is written twice in two different versions.

We again thank the reviewer for realizing our mistake, which we have now fixed.

12. Honestly, I am confused about the LFP / sLFP. I understand that LFP is in the dataset. Please state it clearly already when you introduce the results related to LFP. Why would you need to use sLFP at all? If you have a real LFP you could crop it and analyze UP/DOWN periods separately, similarly as you do with sLFP. If you don’t have the real LFP, what do you present in the figure then? Could you clarify it?

We thank the reviewer for his insightful comment. We now clearly state that the LFPs are already present in the dataset and motivate better the use of sLFPs (page 6). In short, we considered the use of synthetic LFPs appropriate since we could generate these signals solely based on the actual spiking activity of putative pyramidal cells, while the recorded LFPs could potentially suffer from contamination from other sources (such as EMG or volume conduction from other areas). In any case, we have also followed the reviewer’s suggestion and compared SWS to cropped SWS periods (Up only) in the real LFPs. This analysis revealed similar results to the sLFPs. Thus, we have rearranged Fig 4D to include these new results (reproduced below). 

13. (In the context of Figure 4) One of the messages of the paper is that discarding DOWN states leads to dynamics resembling REM / Wake. In order to prove this thesis one would need to perform a statistical test on two samples: one from only UP states and the other from REM and/or Wake, whereas I only find a test comparing DOWN+UP with only UP. From this test solely one can not draw the above mentioned conclusions.

We thank the reviewer for his important remark. We now report the comparison between Wake vs. SWS (UP-only) and REM vs. SWS (UP-only) on Panel 4D and Tables S3 and S4 (please see also our answer to the previous point). Notably, we find that the complexity of SWS UP-only states did not differ from Wake in sLFPs for either of the 3 metrics. The same result held for real LFPs except for the LZ metric: SWS UP-only states showed an even larger complexity than Wake. Furthermore, REM was not different from SWS UP-only states for the sLFPs, while, for real LFPs, REM complexity according to SE and PE was larger than during SWS UP-only states, and the opposite occurred for LZ. Thus, we can conclude that excluding DOWN states leads to complexity levels similar to Wake/REM for sLFPs, and that though there are statistical differences for the real LFPs, we believe using the wording “comparable” is still justified given that the complexity values are much closer to Wake/REM than to the full SWS. 

14. There is a space bar missing at “Accordingly, the sLFP…”.

We thank the reviewer for spotting this error. We have included a space in that location. 

15. Figure 5: is the sigma-nu-z index well written? I am not familiar with it, but it seems a bit puzzling that three symbols are used for one quantity. Perhaps some lower/upper indexing is lost?

We followed the terms employed in Sethna, J., Dahmen, K. & Myers., Nature 2001 and Fontenele et al., PRL 2019. The following plot (taken from Fontenele) shows the use of the exponent, which results from the product between 3 quantities: σ, ν, and z. Briefly, the average size of an avalanche of duration T is given by〈S〉(T) = SoTa, where a = 1/σνz. As Sethna 2001 described, these exponents arise from 1) the cutoff in the avalanche size distribution gets larger following (R−Rc)1/σ when getting closer to the critical point. 2) The typical length L of the largest avalanche is proportional to (R−Rc)ν. 3) The duration T of an avalanche of spatial extent L scales as Lz.

16. Avalanches discussion: I would be curious if a DOWN state can occur during an avalanche, thus cutting it. On one hand it is claimed in the manuscript that the avalanches are the same during UP and REM / Wake, but then in figure 5 B we see that for SWS avalanches tend to be smaller and shorter (the orange line is the lowest) which perhaps could be explained by the cutting effect.

Indeed, the reviewer is right. We have now added a sentence to point to that fact (pg 7). We note, however, that while SWS avalanches are smaller (DOWN states cutting effect), their number of spikes still obeys the criticality relationship.

17. “suggesting that criticality allows for optimal transitions between states.” - how is this claim supported by the results? They only show that avalanches are present and that the model set close to criticality can reproduce them. But transitions between UP/DOWN are imposed onto the model by silencing some stochastic inputs, thus I do not see how are these transitions facilitated by criticality.

We agree with the reviewer’s comment. We softened that sentence, which now reads: “Finally, we reproduce these experimental results by numerical experiments employing a critical branching model, suggesting that criticality may favor transitions between states”

18. In methods silicon probes are mentioned, they record LFP. But not EcoG stripes are mentioned, how were these data collected then?

We thank the reviewer for his question. We have added a summary of the ECoG methods within the Supplementary Methods section.

19. I think tables S3 and S4 should be referring to figures 4, not 3.

We have fixed this mistake. Once again, we thank very much the reviewer for all the effort made during the revision and especially for his keen attention to details. 

Reviewer #2: In this paper, the authors analyze in-vivo recordings from frontal cortical areas and from dorsal CA1 of rats, containing neuronal activity during wakefulness, slow-wave sleep (SWS) and rapid-eye-movement sleep (REM). The authors characterize the dynamics in the phase-space of neuron’s activity by quantifying the amount of recurrence. The authors found that SWS have less complex dynamics compared to awake and REM states, and that this difference can be explained by the presence of DOWN states. Next, they complete this study with analysis on local field potentials (LFPs) and spike avalanches. Finally, they propose a critical branching model to explain their results. The results are interesting and the structure of the manuscript is overall clear. An important point of the manuscript is the link between the level of complexity observed at the macroscopic level, and the differences in the neuronal activity during each state. However, I find that some of the main findings are poorly described, and parts of the text are difficult to understand. I describe here my points of concern.

We thank the reviewer for his work on revising our manuscript. 

Major comments:

(1) The results in Figure 1 are poorly described. The authors used several different measures of complexity, and I believe the authors should improve the description of these measures, and don’t list them twice in the text. Together with this, I suggest the authors should improve the description of the possible recurrence plots in Panel C, and justify why these artificial examples are important to understand the following results.

We thank the reviewer for his comment. We have now updated Figure 1 according to the comments made by both reviewers. Specifically, to make the message clearer, we now only show a 3-neuron example and improve the depiction of how phase-space activity translates into the recurrence plot; moreover, we now also show how the RQA metrics capture the dynamics of the three model examples. The new version of Figure 1 led to related changes in the first subsection of Results, which we hope now improves the understanding of the RQA metrics that are subsequently employed in actual electrophysiological data.

(2) In general, the text in the results section can be improved. For example, in page 5: the last two paragraph are repeated, with few words being replaced. The authors should decide which version to keep. Another example, in page 3: the sentence “An attractor is evidenced as a manifold that attracts different trajectories of the system to the same region of the phase-space; the more convoluted (fractal) the attractor is, the higher the temporal complexity of its trajectories” and the following sentence should be moved somewhere else. If it is there to justify the use of RQA, then I believe there is no need for it.

We thank the reviewer for his comment. We have fixed the paragraph repetition mistake. Moreover, we have improved the text in that paragraph to better exemplify how the RQA metrics work. It now reads: “To illustrate how the RQA metrics behave, we compute them for the examples of Fig. 1C. RR is slightly larger for the periodic system since it recurs more often into similar states than the other examples, while the chaotic trajectory recurs more than the random example. DET and LAM, on the other hand, are maximal for periodic and 

chaotic systems because all points form vertical and diagonal structures, while these 

drop near zero for the random system since recurrent times are rarely connected. 

Moreover, TT is larger for the periodic system since there are no isolated recurrent 

times (all points form small vertical structures). TT decreases in the chaotic system due 

to isolated recurrent times and lowers even further for the random system because 

recurrences occur by chance and rarely form any vertical structure. Finally, DIV is the 

largest for the random system since no diagonal structures are formed, while DIV 

plummets to near zero for the periodic system since all points form long diagonal lines. 

DIV lies in-between for the chaotic system since it forms short diagonal lines. Thus, 

predictability in the system trajectory is quantified by RR, DET, LAM, and TT, where 

the larger [smaller] their values, the more [less] predictable. On the other hand, 

randomness is quantified by DIV, where the larger [smaller] its value, the more 

divergent [convergent] the trajectory”

(3) In Figure 4, I don’t understand the need to use surrogate LFP instead of just using the LFP from the data. The authors first justify the use of sLFP as “The motivation behind this method is that it allows to precisely control the sources which dictate the field potential and avoid the influence of any external variable not directly related to spiking activity”. Then, they show in panel C how the sLFP is similar to the LFP, and in panel D that the entropy methods they used lead to very similar results when applied to either LFP or sLFP. This, to me, suggest that there is no real difference between the two types of signals in terms of complexity. However, the results in panel E support the main claim of the paper, but they are shown only for the sLFP. It is important to see that these results hold true also for the LFP data. I suggest the authors to apply the analysis of panel E on the raw LFP, to move the results on the sLFP in the supplementary, and to keep the results on the raw data LFP as the main point. Otherwise, give an explanation on why the raw LFP cannot be analysed as in panel E.

We thank the reviewer for his insightful comment. In the revised version, we have followed his suggestion and compared SWS to SWS Up only states in the real LFPs as well, obtaining similar results to the sLFPs. We have now rearranged Fig 4D to include these new results. We have also expanded our motivation for the use of sLFPs (page 6). Please see also our answer to Reviewer 1, who also raised this issue.

(4) The authors referred to some supplementary figures (Figure S4, S5 and S6) only in the Discussion section. I believe the authors should refer to those figures in the Result section.

We thank the reviewer for his comment. We have now added several phrases to reference those figures inside the Results section. Please note that because these figures are referenced earlier now, their numbering changed (S4 to S3, S5 to S6, and S6 to S2).

Minor comments:

(1) I find that all the figures of the paper are too small compared to the text. One needs to pretty much zoom in to be able to see and understand the results. I am not aware if this problem can be solved during formatting. I suggest that the authors update the figures to increase their readability.

We thank the reviewer for this comment, which we agree. We believe that the figures will be larger in the final PDF version of our manuscript (so as to extend beyond the text margins). In any case, we have increased the font size in several of the figures.

(2) I believe that the average reader is not necessarily familiar with the concept of “complexity”. I suggest the author to better define the notion of the complexity of a neuronal signal in the text, especially the abstract and the introduction.

We thank the reviewer for pointing out that the average reader might not be familiar with the notion of complexity in a neuronal signal. We have thus modified sentences introducing the term in both the abstract and introduction.

(3) Page 5: “The square-shaped recurrences appearing during SWS in Fig.2 can be generated by two possible mechanisms. Either the neurons remain constantly active for a period of time, or they remain silent (null firing counts) and give rise to a trajectory in the origin of the phase space. Next, we show that the latter is true and is mainly due to DOWN states“. I do not understand why a squared-shaped recurrence could not be given by some neuron firing and some remaining silent, as along as they keep this activity for a period of time.

We agree with the reviewer’s comment. We modified the sentence, which now reads: ¨Either a subset of neurons (or even all) remains constantly active for a period of time, or the neurons remain silent (null firing counts) corresponding to a trajectory in the origin of the phase space¨. 

(4) Figure 6: I suggest the authors should show also some metrics of variation around the mean, and consider doing a statistical analysis to quantify the significance of the difference in the models.

We thank the reviewer for raising this point. We have now included the standard deviation along with the mean in Figure 6C. Please note that since these results are derived from simulations, if we were to plot SEMs (instead of SDs) they could be made arbitrarily small (outside the critical point), given that the sample size (i.e., number of simulations) can be extended as much as one wished. By the same token, all visible mean differences reach statistical significance (with p value ->0) with the number of simulations, making the use of statistical tests less relevant when compared to the report of experimental results.

---

## [Decision Letter · Decision Letter 1]

3 Aug 2023

Sleep disrupts complex spiking dynamics in the neocortex and hippocampus

PONE-D-23-12688R1

Dear Dr. Gonzalez,

We’re pleased to inform you that your manuscript has been judged scientifically suitable for publication and will be formally accepted for publication once it meets all outstanding technical requirements.

Kind regards,

Jordi Garcia-Ojalvo

Academic Editor

PLOS ONE

Additional Editor Comments (optional):

Reviewers' comments:

Reviewer's Responses to Questions

**Comments to the Author**

1. If the authors have adequately addressed your comments raised in a previous round of review and you feel that this manuscript is now acceptable for publication, you may indicate that here to bypass the “Comments to the Author” section, enter your conflict of interest statement in the “Confidential to Editor” section, and submit your "Accept" recommendation.

Reviewer #1: All comments have been addressed

Reviewer #2: All comments have been addressed

2. Is the manuscript technically sound, and do the data support the conclusions?

Reviewer #1: Yes

Reviewer #2: Partly

3. Has the statistical analysis been performed appropriately and rigorously? 

Reviewer #1: Yes

Reviewer #2: Yes

4. Have the authors made all data underlying the findings in their manuscript fully available?

Reviewer #1: Yes

Reviewer #2: Yes

5. Is the manuscript presented in an intelligible fashion and written in standard English?

Reviewer #1: Yes

Reviewer #2: Yes

6. Review Comments to the Author

Reviewer #1: In the added description explaining the methodology ti, tj, tl and tk times don't have lower indeces.

I don't know if applying them was the authors' intention.

Reviewer #2: (No Response)

7. PLOS authors have the option to publish the peer review history of their article (what does this mean?). If published, this will include your full peer review and any attached files.

Reviewer #1: **Yes: **Maciej Jedynak

Reviewer #2: **Yes: **Matteo Saponati

---

## [Editor Report · Acceptance letter]

7 Aug 2023

PONE-D-23-12688R1 

Sleep disrupts complex spiking dynamics in the neocortex and hippocampus 

Dear Dr. González:

I'm pleased to inform you that your manuscript has been deemed suitable for publication in PLOS ONE. Congratulations! Your manuscript is now with our production department. 

Kind regards, 

on behalf of

Dr. Jordi Garcia-Ojalvo 

Academic Editor

PLOS ONE